# Multiplex PCR Based Strategy for Detection of Fungal Pathogen DNA in Patients with Suspected Invasive Fungal Infections

**DOI:** 10.3390/jof6040308

**Published:** 2020-11-23

**Authors:** Joana Carvalho-Pereira, Filipa Fernandes, Ricardo Araújo, Jan Springer, Juergen Loeffler, María José Buitrago, Célia Pais, Paula Sampaio

**Affiliations:** 1Centre of Molecular and Environmental Biology (CBMA), Department of Biology, University of Minho, 4710 Braga, Portugal; joanapereira@bio.uminho.pt (J.C.-P.); filipafernandes_11@hotmail.com (F.F.); cpais@bio.uminho.pt (C.P.); 2Department Medical Biotechnology, Health Sciences, Flinders University, Adelaide 5042, Australia; ricardo.pintoaraujo@flinders.edu.au; 3Department of Internal Medicine II, University Hospital of Würzburg, 97080 Würzburg, Germany; SPRINGER_J@ukw.de (J.S.); Loeffler_J@ukw.de (J.L.); 4Mycology Reference Laboratory, National Centre of Microbiology, Instituto de Salud Carlos III, 28220 Madrid, Spain; buitrago@isciii.es

**Keywords:** molecular diagnosis, fungal infections, multiplex PCR, *Candida* sp., *Aspergillus* sp., *Rhizopus arrhizus*

## Abstract

A new and easy polymerase chain reaction (PCR) multiplex strategy, for the identification of the most common fungal species involved in invasive fungal infections (IFI) was developed in this work. Two panels with species-specific markers were designed, the *Candida Panel* for the identification of *Candida* species, and the *Filamentous Fungi Panel* for the identification of *Aspergillus* species and *Rhizopus*
*arrhizus*. The method allowed the correct identification of all targeted pathogens using extracted DNA or by colony PCR, showed no cross-reactivity with nontargeted species and allowed identification of different species in mixed infections. Sensitivity reached 10 to 1 pg of DNA and was suitable for clinical samples from sterile sites, with a sensitivity of 89% and specificity of 100%. Overall, the study showed that the new method is suitable for the identification of the ten most important fungal species involved in IFI, not only from positive blood cultures but also from clinical samples from sterile sites. The method provides a unique characteristic, of seeing the peak in the specific region of the panel with the correct fluorescence dye, that aids the ruling out of unspecific amplifications. Furthermore, the panels can be further customized, selecting markers for different species and/or resistance genes.

## 1. Introduction

The development of medicine, contributing to a prolonged lifespan and a better quality of life, has also caused an increase in the number of immunocompromised individuals with predisposing risk factors for infectious disease. Cancer chemotherapy, organ transplantation, antimicrobial therapy and abdominal surgery are among the medical procedures that constitute the main risk factors predisposing to invasive fungal infections (IFI) [1,2]. IFI are associated with mortality rates, ranging from 2% to 60%, depending on the etiology of invasive infection and the type of underlying disease [3], with high hospital costs [4,5]. A recent analysis estimated that in the United States the total direct medical cost of fungal-disease related hospitalizations was $4.6 billion [5]. This analysis also indicated $1.4 billion of this sum to be due to *Candida* infections (26,735 hospitalizations,) and $1.2 billion to *Aspergillus* infections (14,820 hospitalizations) [5]. Notably, this study also reported that the analysis did not account for the costs related to unnecessary medical procedures and inappropriate treatment before the fungal diagnosis was established, so medical costs would have been much higher than $4.6 billion.

Several etiological agents are associated with IFI but species of the genus *Candida*, closely followed by *Aspergillus* sp., are still the most frequently isolated [6]. Although *Candida albicans* is most often isolated, other *Candida* species such as *C. tropicalis, C. glabrata, C. parapsilosis* and *C. krusei* are also increasingly recognized as widespread pathogens [7]. Concerning *Aspergillus* sp., *A. fumigatus* is the dominant species, *A. flavus*, *A. niger* and *A. terreus* also being largely associated with invasive aspergillosis in severely immunocompromised individuals [8]. More recently, outbreaks of infection caused by Mucorales (mucormycosis) associated with IFI were reported in developed countries [6,9]. Mucorales are considered emerging pathogenic species, particularly *Rhizopus, Lichtheimia* and *Mucor* genus, *Rhizopus arrhizus* (former *Rhizopus oryzae)* being the most frequently found [10].

An early and adequate identification of the infecting species is crucial since it allows the selection of appropriate antifungal treatment, enhancing survival of infected patients [11,12]. Moreover, since the severity of clinical manifestations and risk of deep organ involvement are species dependent, the significance of an early and correct diagnosis is even higher [2]. Species identification is also essential for the rapid identification and control of hospital outbreaks [13].

The European Organization for Research and Treatment of Cancer/Invasive Fungal Infections Cooperative Group and the National Institute of Allergy and Infectious Diseases Mycoses Study Group (EORTC/MSG) states that, for proven diagnosis of IFI, a microbiological and/or histopathological diagnostic method is required [14]. Thus, blood cultures are still the reference standard for the microbiological diagnosis of IFI [15]. However, this culture-based method has several limitations, such as false-negative results because of on-going antimicrobial therapy, and long time to positivity (from 12 h to 72 h). These drawbacks reflect the low numbers of circulating microorganisms (from 1 to 10 CFU/mL), the possible presence of fastidious pathogens, and insufficient blood sample volumes (especially in children). Therefore, their sensitivity is low, only 50–75%, even when sampling recommendations are correctly followed [16]. Bacteremia coexists in approximately 20% of cases, hampering fungi detection even further [16]. However, blood cultures are relatively inexpensive, require readily available technologies and allow the evaluation of anti-microbial pathogen susceptibility.

To overcome the limitations of blood cultures and achieve a faster diagnosis, laboratory techniques that do not require microorganism culture growth or its identification in tissue samples play an important role in clinical decision-making related to IFI [17]. Several methods have been developed to detect fungi antigens in various types of sample, such as positive blood culture specimens, whole blood, plasma, serum or bronchoalveolar lavages. Detection of the fungal cell wall component (1, 3)-β-D-glucan (BDG) is commonly performed as a screening and diagnostic tool for patients with suspected IFI [18,19]. BDG is a pan-fungal antigen identified by several assays, including Fungitell Assay (Associates of Cape Code Inc., Falmouth, MA, USA) approved by the Food and Drug Administration (FDA) to test serum of patients suspected of infection by *Candida* spp., *Pneumocystis jiroveci, Aspergillus* spp., *Fusarium* spp., and *Acremonium* spp. [20,21,22]. Another FDA approved assay, Platelia *Aspergillus* EIA (Bio-Rad, Marnes-la-Coquette, France), was designed to detect galactomannan (GM) in the serum and bronchoalveolar lavages of suspected patients. GM is produced by a variety of fungi, including *Aspergillus* spp., *Penicillium* spp., *Paecilomyces* spp., *Purpureocillium licacinum*, and *Histoplasma* spp. Laser desorption/ionization time-of-flight mass spectrometry (MALDI-TOF MS) analyses have been recently used for microbial identification and diagnostics [23]. MALDI-TOF MS has proven to be effective for the identification of yeast species, including *Candida* [24]. However, for filamentous fungi identification, there have been challenges, such as the lack of a standardized process for culturing and extracting proteins for the analysis and limited spectral libraries.

Molecular methodologies, based on the analysis of DNA, are characterized by their high specificity, sensibility and reproducibility. The majority of DNA based techniques require DNA detection by singleplex or multiplex polymerase chain reaction (PCR) amplification, using species-specific primers. The PCR fragments can then be identified by several methods, including sequencing, melting curve analysis, electrophoretic separation, fluorescence in situ hybridization (FISH), electrospray ionization mass spectrometry (ESI-MS), microarray, or surface-enhanced Raman scattering (SERS) [25]. SeptiFast (Roche Diagnostics) and MycAssay *Aspergillus* kits are some examples of commercial kits used in clinical laboratories to identify fungal pathogens by real-time PCR. These kits do not require prior fungal culture and the amplification is performed directly from clinical samples [26,27]. Other multiplex assay such as the MycoReal Candida PCR test (Ingenetix) [28] and the Fungiplex Candida IVD PCR Kit (Bruker Daltonik) [29] are also available but they require extraction of DNA from whole blood, serum or plasma. Several other multiplex PCR methods have been described in the literature and described in reviews [30]. These methodologies usually use nonspecific targets (e.g., mitochondrial DNA, 28S rRNA, 18S rRNA, and internal transcribed spacer (ITS) regions of ribosome DNA) with higher probability of nonspecific signals [31,32]. In mixed microbial infections, the overlapping of the molecular targets may also interfere with the correct identification [33].

In this work, we designed a sensitive and specific multiplex PCR strategy for the identification of the most clinically important pathogenic species of *Candida* spp., *Aspergillus* spp. and *Rhizopus arrhizus*. Two multiplex panels were tested and optimized, namely, the *Candida Panel*, using primers specific for *C. albicans, C. parapsilosis, C. glabrata, C. krusei, C. tropicalis* and the *Filamentous Fungi Panel*, using primer pairs designed for identification of *A. fumigatus, A. flavus, A. terreus, A. niger* and *R. arrhizus*. The novelty of this method is that it uses species specific primers outside the mitochondrial or ribosomal DNA, reducing cross-amplification with closely related species, and that the visualization of the results is straightforward, based on the presence of fragments with the correct size and fluorescent color, ruling-out unspecific amplifications. As a proof-of-concept, this methodology was tested using DNA isolated from clinical sample strains and with DNA isolated directly from clinical samples from biopsies, bronchoalveolar lavages, cerebrospinal fluid, and vitreous humor.

## 2. Materials and Methods

### 2.1. Primer Selection and Panel Design

Specific primers were designed by using Primer 3Plus (https://primer3plus.com) software for selected DNA regions. Reverse primers were fluorescently labelled with HEX (hexa-chloro-fluorescein), FAM (6-carboxyfluorescein) or NED (Proprietary to Applied Biosystems, Califórnia, USA) for DNA size determination in 3130 Genetic Analyzer (Applied Biosystems, Califórnia, USA) (Table 1). By combining the expected fragment sizes with the fluorescent label, two panels were designed combining the sizes with the fluorescent label.

### 2.2. Strains Tested

DNA from 121 previously identified clinical and environmental fungal isolates was used to optimize the methodology. To assess specificity of the methodology, strains of the following species were used: *C. albicans, C. parapsilosis sensu stricto*, *C. glabrata sensu stricto, C. tropicalis, C. krusei, C. metapsilosis, C. orthopsilosis, C. bracarensis, C. guilliermondii, C. lusitaniae, Lodderomyces elongisporus, A. fumigatus, A. flavus, A. terreus, A. niger, A. versicolor, A. unguis, A. westerdijkiae, A. tamarii, A. wentii, A. nidulans, A. tubingensis, R. arrhizus, R. microsporus, Mucor* sp. and *Lichtheimia corymbifera*, recovered from different sources such as saliva, vagina, respiratory tract, feces, skin, blood culture, urine, catheter, wine and hospital environment. All yeast strains were isolated from clinical samples except for *L. elongisporus* from a wine sample, while 63% of the filamentous fungi were isolated from clinical samples and 37% from the hospital environment. All these isolates are conserved at the collection of the Biology Department of Minho University, Braga, Portugal. The type strains, *C. albicans* (PYCC 3436), *C. tropicalis* (PYCC 3097), *C. krusei* (PYCC 3341), *C. glabrata* (PYCC 2418) and *C. bracarensis* (153MT), obtained from the Portuguese Yeast Culture Collection (PYCC), New University of Lisbon, Lisbon, Portugal, were also included in this study. Human Genomic DNA (Promega) was also used in the development of panels to assess cross-amplification.

### 2.3. Genomic DNA Extraction

Before DNA extraction, yeasts and filamentous strains were grown at 30 °C for 48 h and five days, respectively, on YPD (Yeast extract Peptone Dextrose - yeast extract 1%, Bacto Peptone 1%, glucose 2% and agar 2%) agar medium.

Yeast DNA was extracted using the commercial kit JetQuick TM DNA Purification Kit (Invitrogen, Carlsbad, CA, USA), according to manufacturer’s instructions. In order to evaluate if the new methodology could be used directly with lysed yeast cells, without previous DNA extraction, colony PCR was performed by using a small amount of cells directly in the PCR tube and the DNA template was obtained by thermal shock, as previously described [34].

Genomic DNA from filamentous fungi was extracted using an in-house protocol. Briefly, fungal biomass was frozen in liquid nitrogen and macerated using a pestle. The maceration step was repeated three times with the addition of TES buffer (0.05 M EDTA, 20% sucrose, 1M Tris pH 8.0). After maceration, samples were heated at 100 °C, centrifuged 10 min at 1000× *g*, 4 °C, and centrifuged again at 19,000× *g*, 3 min, 4 °C. Chloroform: isoamyl alcohol (24:1) was added and samples centrifuged at 20,000× *g* for 10 min at 4 °C. The top layer was transferred to a new 2 mL tube and 1/10 of the total volume of sodium acetate 3 M and 2.5 × the total volume of absolute ethanol was added. Samples were centrifuged over 30 min at 20,000 rpm. The supernatant was discarded and the pellet was washed with 70% (*v/v*) ethanol. Samples were washed again with 70% (*v/v*) ethanol and pellets dried at room temperature. Finally, the pellets were resuspended in ultrapure water and heated at 65 °C for 45 min.

### 2.4. DNA Extraction from Serum Spiked with DNA

Clotted-blood samples were collected from consenting healthy donors, and serum fractionated by centrifugation was divided into 500 µL aliquots. Then, serum aliquots were spiked with 5 ng, 500 pg and 50 pg of DNA from *A. fumigatus, A. terreus, C. albicans, C. krusei* or *R. arrhizus* and total DNA was extracted using the QIAmp Ultrasense Virus Kit (QIAGEN, Hilden, Germany) and eluted with 20µL of buffer, according to the European Aspergillus PCR Initiative (EAPCRI) for testing serum [35]. The DNA obtained was stored at −20 °C for amplification. The efficiency of extraction was determined comparing qPCR results (Springer et al. (2017) [36]) using DNA before vs. DNA after QIAmp Ultrasense Virus Kit extraction. All manipulations were performed in a category 2 laminar flow cabinet to prevent contamination.

### 2.5. PCR Amplification Conditions

For all primer pairs, singleplex PCRs were performed using several strains, to confirm locus-specific amplification and PCR fragment sizes. All loci were amplified with 25 ng of genomic DNA, in a 10 µL reaction volume, containing 1 × PCR buffer (20 mM Tris HCl [pH 8.4], 50 mM KCl), 0.2 mM of each of the four deoxy-nucleoside triphosphates (dNTPs), 1.5 mM MgCl_2_, 1U of PhusionTM High-Fidelity DNA Polymerase (Thermo Scientific, Massachusetts, USA), 0.2 µM of each primer, and 2 µL of fungal DNA. The PCR program consisted of a pre-incubation step for 5 min at 95 °C, followed by 30 cycles of 30 s at 94 °C, 30 s at 64 °C and 1 min at 72 °C, with a final extension step of 10 min at 72 °C.

Multiplex PCRs were performed by combining 1 × PCR buffer, 0.2 mM of each of the four deoxy-nucleoside triphosphates (dNTPs), and 2 mM MgCl_2_, 2U of PlatinumTM II *Taq* Hot-Start DNA polymerase (Invitrogen, Califórnia, USA) in a 20 µL total volume. The PCR program consisted of a pre-incubation for 5 min at 95 °C, followed by 45 cycles of denaturation 30 s at 95 °C, annealing for 30 s at 60 °C and extension for 1 min at 72 °C, with a final extension of 10 min at 72 °C. The primer concentration used was adjusted according to the PCR panel used.

### 2.6. PCR Fragment Size Determination

Following amplification, PCR products were added to a mixture (9:1) of Hi-Di formamide and internal size standard (GeneScan 500 6-carboxy-X-rhodamine [ROX], Applied Biosystems Inc., Califórnia, USA), and PCR fragments were separated in 3130/3130xl Genetic Analyzers (Applied Biosystems, Califórnia, USA). Fragment sizes of the PCR products were determined automatically using the Peak Scanner software (www.thermofisher.com).

### 2.7. Clinical Samples

This study was retrospective and noninterventional. A total of 29 clinical samples, obtained from the DNA collection of clinical samples from the Instituto de Salud Carlos III, Centro Nacional de Microbiología (Madrid, Spain) previously anonymized in compliance with Spanish law, and Medizinische Klinik II, Universitätklinikum Würzburg (Würzburg, Germany), were tested using the multiplex PCR panels. These samples included 19 samples of patients with probable/proven fungal infection and 10 negative controls, and were obtained from biopsies, bronchoalveolar lavages (BAL), cerebrospinal fluid, and vitreous humor.

## 3. Results

### 3.1. Design of the Multiplex Strategy

The search for specific regions of *Candida, Aspergillus* and *Rhizopus* species allowed the design of ten specific primer pairs with annealing temperatures around 60 °C, to ensure specificity and reproducibility. The differences in the molecular weight of the expected amplified PCR fragments and the possibility of combining different fluorescent dyes allowed the design of two panels to identify different sets of species, namely, the *Candida Panel*, for identification of *C. albicans, C. parapsilosis, C. glabrata, C. krusei* and *C. tropicalis* (Figure 1a), and the *Filamentous Fungi Panel*, for identification of *A. fumigatus, A. flavus, A. terreus, A. niger* and *R. arrhizus* (Figure 1b).

All primers were first tested in singleplex PCR, using different strains to confirm their specificity. The amplification was successful for all markers, and the fragment sizes were within the expected size (Figure 2).

These markers presented 100% specificity since no cross-amplification products were obtained when primers and PCR conditions were used to amplify other species (Table 2). The specificity regarding phylogenetical related or frequently misidentified species was noteworthy, i.e., *Candida parapsilosis, C. orthopsilosis* and *C. metapsilosis,* or *R. arrhizus*, *R. microsporus* and *L. corymbifera* (Appendix A).

### 3.2. Optimization of Multiplex Amplification Conditions

Amplification conditions for multiplex PCR were optimized and results were compared with those obtained in the singleplex analyses. Different concentrations of MgCl_2_ (from 1.5 to 2.5 mM), primer concentration (from 0.11 to 0.25 µM of each primer) and annealing temperatures (from 55 °C to 64 °C) were tested. Results showed that in multiplex PCR MgCl_2_ 2.0 mM promoted the best amplification profile, but no significant differences were observed when using different primer concentrations [37]. All annealing temperatures ensured the presence of the expected PCR products; however, at 60 °C a balance between the peak intensity of the expected fragment size and reduced unspecific peaks was observed (Appendix A).

PCR profile of the same strains analyzed in Figure 2, were obtained using 2 mM MgCl_2_, 0.2 µM of each primer, and annealing temperature of 60 °C (Figure 3). Comparing both panels, some differences were observed regarding the intensity of the peaks and the presence/absence of a second PCR fragment (allele). These differences do not compromise the correct identification of the different species.

The same multiplex PCR conditions were also tested using colony PCR showing similar results [37] which indicates that the multiplex PCR panels could be used for rapid identification of positive blood-cultures, either by using DNA extracted from the positive fungal cultures or directly using the fungal cells in the PCR.

### 3.3. Limit of Detection

In clinical practice, low amounts of fungal DNA are present in clinical samples. Therefore, the number of PCR amplification cycles, namely 35, 40, 45 and 50, was tested to enhance the sensitivity of the methodology, using serial dilutions of fungal DNA (from 10 ng to 1 pg of DNA present in the PCR tube). Table 3 presents the results observed for all species tested. DNA from all species amplified very well with high mounts of DNA at all PCR cycles. However, with the lowest DNA amounts tested, 1 pg, not all species amplified, and those that amplified needed 40 or more PCR cycles, with 10 pg of fungal DNA amplification observed only with 45 and 50 amplification cycles and, since no significant differences were registered, amplifications with 45 cycles were selected.

Figure 4 presents amplification profiles using 100 pg to 1 pg of DNA from *C. glabrata* and *A. flavus*, using the corresponding panels. No amplification peaks were observed in electropherogram analysis with the *Candida Panel* or with the *Filamentous Fungi Panel* when both panels were tested with human DNA (Figure 4).

### 3.4. Identification of Mixed Fungal DNA

The ability of the multiplex strategy to identify the different species in mixed infections was also evaluated. Two combinations, DNA of *C. albicans* and *C. tropicalis,* and DNA of *A. fumigatus* and *A. niger,* were deliberately mixed in the same tube and multiplex PCR using the respective fungal panels was performed as previously described. In both combinations, two amplification profiles were obtained consistent with the two different species mixed (Figure 5).

### 3.5. Human Serum Spiked with Fungal DNA

To mimic clinical samples, 500 µL of serum from healthy donors was spiked with 5 ng, 500 pg and 50 pg of fungal DNA (*A. fumigatus*, *A. terreus*, *C. albicans*, *C. krusei* or *R. arrhizus)*, and analysed with the corresponding multiplex panels. The calculated yield of extraction was approximately 76.5%; therefore, the DNA final concentrations tested ranged from 190 pg/µL to 1.91 pg/µL of fungal DNA. Results obtained showed the presence of the expected peak in the electropherogram, indicating the correct species identification. Figure 6 shows the amplification profile of serum spiked with 500 pg of fungal DNA.

### 3.6. Identification of Fungal DNA in Clinical Samples

To determine if the methodology can detect fungal DNA from clinical samples from patients with IFI, total DNA was extracted from 29 clinical samples, 19 samples from patients with probable invasive fungal infection and ten samples as negative controls from patients with other respiratory infections. Samples tested were obtained by biopsy (*n* = 11), 13 bronchoalveolar lavages/broncho-aspirates (*n* = 13), cerebrospinal fluids (*n* = 3), whole blood (*n* = 1) and vitreous humor (*n* = 1). A total of 29 clinical samples were tested using our methodology (Table 4). DNA was extracted using a Qiamp DNA mini-kit (Qiagen) according to manufacturer’s instructions in a 50 µL elution volume. Two microliters of this DNA were used for the assay.

Results obtained indicated no amplification profile in all samples from patients without fungal infections (CNM20 to CNM24). Excluding controls, from the 19 clinical samples from patients with *Candida* or *Aspergillus* infections, ten samples confirmed the respective infection. Figure 7 shows the amplification profile of DNA extracted from samples CNM5 and CNM9, with infections due to *C. albicans* and *A. fumigatus,* respectively.

Some problems with samples from bronchoalveolar lavages and broncho-aspirates were observed. Considering the 10 clinical samples, excluding controls, three did not amplify at all, five were classified as inconclusive and two amplified as expected. In the inconclusive samples the expected PCR peak was present; however, the high background hindered interpretation (Appendix A).

Although the number of clinical samples was low, we calculated the sensitivity and specificity of the methodology. Considering all clinical samples, the methodology presented 52 % sensitivity and 100% specificity. However, if we consider only samples from sterile sites (biopsies, cerebrospinal fluids and vitreous humor), the sensitivity of the methodology was 89%. Considering the different panels, the overall sensitivity for *Candida Panel* was 82% and specificity 100%, but considering samples from sterile places sensitivity raised to 88%. These values were not calculated for the *Filamentous Fungi Panel,* because the majority of the samples were from bronchoalveolar lavages and broncho-aspirates, which were the problematic samples.

## 4. Discussion

The rapid and accurate identification of pathogens responsible for invasive fungal infections is of extreme relevance for a prompt diagnosis and selection of a correct therapeutic strategy. The identification of these species remains challenging since the conventional methods, such as biopsy and culture, are time-consuming and show low sensitivity and specificity. A way of improving the clinical utility and performance of current fungal diagnostics is by interpreting the results of two individual tests used in combination, such as combining molecular and antigen testing results. The approach applied most frequently is the combination of GM testing and *Aspergillus* polymerase chain reaction (PCR) diagnostics [38].

In this study, we designed a new PCR-based methodology that can also be used in combination with other diagnostic tests. The main advantages of our methodology, compared to other PCR based methods, are the ability to identify the ten most frequently isolated fungal species and species in mixed infections, and the ease of interpreting the results. This last characteristic is due to the possibility of the user effectively seeing the peak in the corresponding panel position (molecular weight) with the corresponding color (fluorescent dye). Another important characteristic of this methodology is the possibility of customization. New markers can be included in the multiplex panels, according to the epidemiology of the specific geographical region.

This work presents a proof-of-concept, demonstrating how this methodology can be designed and applied. The DNA fragment sizes are crucial in planning the multiplex strategy since differences in the size of the alleles amplified at each of the selected loci must be combined with different fluorescent dyes to make simultaneous amplification possible. All the primer pairs in the multiplex PCR should also enable similar amplification efficiencies for their respective target. Thus, in this work, primers were carefully selected and designed to compose the desired panels. This design implicated the knowledge of the expected fragment sizes and the combination of different fluorescent dyes when the expected sizes were similar (Figure 1). The windows of the designed panels are not saturated, and other markers (new species or resistance genes) can be included.

The methodology was tested in singleplex and in multiplex PCR conditions. We observed that in multiplex PCR the intensity of some peaks was lower, and the presence of a second allele was absent in some samples. These differences could be due to a lower PCR efficiency in multiplex, but the general species identifications were still clear.

The specificity, reproducibility and sensitivity of this multiplex assay were evaluated first in analytical tests with DNA from different fungal species. Results showed that the proposed method detected correctly all ten targeted pathogens, exhibited no cross-reactivity with nontargeted species and presented a limit of detection of 10 to 1 pg of DNA. The limit of detection in PCR assays is highly dependent on the method used to detect the amplified fragments, being much lower for pPCR and GeneScan analysis, in the range of pg to fg of DNA, compared with conventional electrophoresis (agarose or polyacrylamide gels), usually ng of DNA [39]. This multiplex panel for filamentous fungi was able to distinguish *Aspergillus* species and *R. arrhizus,* which has been a problem in other multiplex strategies [40]. The fact that this methodology can distinguish the different pathogens in mixed infections is a clear advantage over other assays. The possibility of the user seeing the peak in the specific region of the panel, along with the correct fluorescence dye, is also a unique characteristic of this assay that aids in ruling out unspecific amplifications.

The ability of the methodology to correctly identify the fungal species in human serum spiked with fungal DNA, without interference with human DNA, was a promising indication. The next step was to test the methodology with clinical samples. Results from the analysis using bronchoalveolar samples showed that the method, as it is, is not suitable for this type of specimen. Bronchoalveolar lavages are heterogeneous samples, as volumes are different in each case and some samples are very diluted, resulting in suboptimal DNA extraction. Other factors affecting the amplification may be the presence of PCR inhibitors or the presence of large quantities of non-target microbial DNA. We used the same DNA extraction method for all clinical samples. In future work, it would be useful to try to concentrate the sample before analysis and exploit other DNA extraction methods to neutralize possible PCR inhibitors [41]. Nonetheless, this methodology was suitable for clinical samples from sterile sites, such as from biopsies, cerebrospinal fluid, and vitreous humor.

Considering all clinical samples, our methodology presented 52% sensitivity and 100% specificity. This sensitivity value may be considered low compared with values observed in the literature for multiplex PCR assays (from 58% to 99.4%) [42]. The sensitivity of the FDA approved T2Candida assay for blood samples ranges from 59% to 99.4% [43]. The multiplex PCR LightCycler SeptiFast test presented a sensitivity of around 85% and a specificity of 93.4% [44]. However, comparisons of these values are affected by several factors, including the type and diversity of clinical samples used in the analysis [42]. The majority of the multiplex PCR studies used blood or serum samples while in our study we used mainly samples from biopsies. In the present study, if we consider only samples from sterile sites the sensitivity reaches 89%, which is within acceptable values. All samples from control patients presented no amplification.

This proof-of-concept study showed the potential of this multiplex PCR methodology to be used for the identification of pathogenic species. We developed panels suitable for the identification of the ten most important fungal species involved in IFI. Similar panels can be developed, not only for fungal species, but also for other clinically important microbial pathogens and genes of interest. If markers are well designed, the methodology shows high stability and capacity to discriminate between the different species, even if more than one species is simultaneously present in the sample. Moreover, the methodology developed is easy to perform and the visual discrimination of the amplified peaks helps to rule out unspecific amplifications and can be implemented at relatively low cost for routine identification in microbiological laboratories, in a short time (approximately 2 h 20 min). If performed correctly, this analysis could also be performed directly from the cell biomass of positive blood cultures, without the need for DNA extraction, in less than two hours. This methodology proved to be suitable for identification of the ten fungal species in the panels, either from positive blood cultures or DNA extracted from sterile sites, but a validation with more samples from patients with proven infection will be needed.

## 5. Patents

The work reported in this manuscript resulted in the patent N° 20181000078137, of Instituto Nacional da Propriedade Industrial (INPI): “Método para identificar microrganismos patogénicos de resistência usando análise de fragmentos de PCR multiplex” 2020.

## Figures and Tables

**Figure 1 jof-06-00308-f001:**
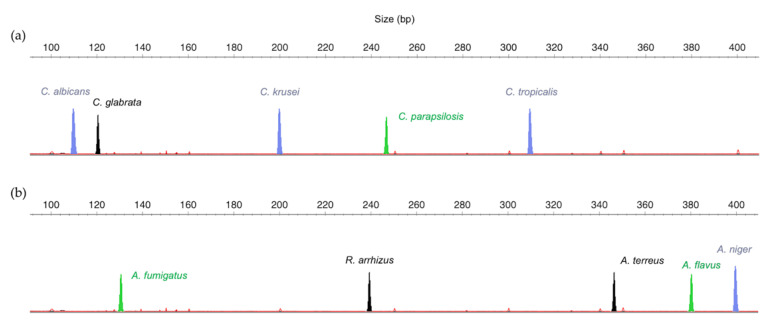
Schematic representation of: (**a**) *Candida Panel* and (**b**) *Filamentous Fungi Panel*, combining the molecular weight of the polymerase chain reaction (PCR) products with different dyes (FAM = blue, HEX = green, NED = black).

**Figure 2 jof-06-00308-f002:**
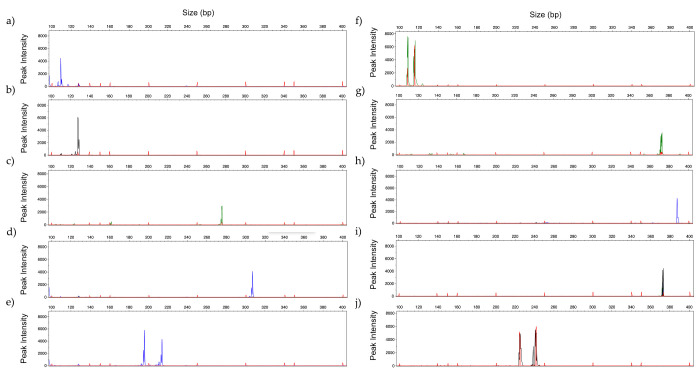
Representative GeneScan profiles obtained by singleplex analysis with 2ng of DNA of: (**a**) *C. albicans (blue);* (**b**) *C. glabrata* (black); (**c**) *C. parapsilosis* (green); (**d**) *C. tropicalis* (blue)*;* (**e**) *C. krusei* (blue)*;* (**f**) *A. fumigatus* (green); (**g**) *A. flavus* (green)*;* (**h**) *A. niger* (blue)*;* (**i**) *A. terreus* (black); and (**j**) *R. arrhizus* (black). (Fluorescent dyes: FAM = blue, HEX = green, NED = black).

**Figure 3 jof-06-00308-f003:**
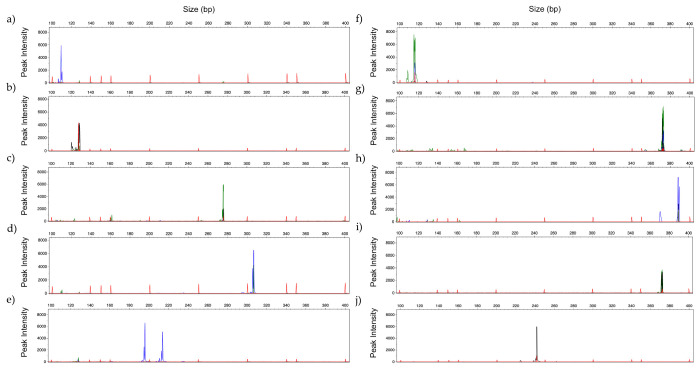
Representative GeneScan profiles obtained by multiplex analysis with 2 ng of DNA using *Candida Panel* of: (**a**) *C. albicans* (blue); (**b**) *C. glabrata* (black); (**c**) *C. parapsilosis* (green); (**d**) *C. tropicalis* (blue) and (**e**) *C. krusei* (blue)*;* and Filamentous Fungi Panel of: (**f**) *A. fumigatus* (green); (**g**) *A. flavus* (green); (**h**) *A. niger* (blue); (**i**) *A. terreus* (black) and (**j**) *R. arrhizus* (black). (Fluorescent dyes FAM = blue, HEX = green, NED = black).

**Figure 4 jof-06-00308-f004:**
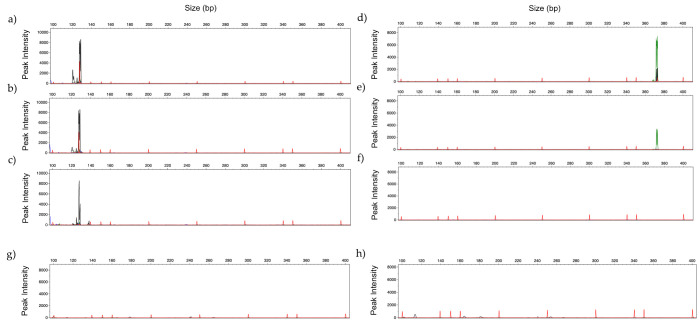
GeneScan profiles obtained by multiplex analysis testing: (**a**) 100 pg; (**b**) 10 pg and (**c**) 1 pg of DNA of *C. glabrata* (black) with *Candida Panel*, and (**d**) 100 pg; (**e**) 10 pg and (**f**) 1 pg of DNA of *A. flavus* (green)with *Filamentous Fungi Panel*. (**g**,**h**) represent the multiplex profile with 486 ng of commercial human DNA, with *Candida Panel* and *Filamentous Fungi Panel*, respectively. (Fluorescent dyes HEX = green, NED = black).

**Figure 5 jof-06-00308-f005:**
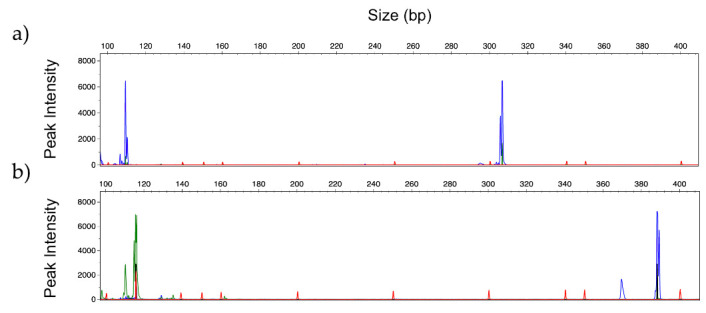
GeneScan profiles obtained by multiplex analysis of 100pg of mixed gDNA of: (**a**) *C. albicans* (blue) and *C. tropicalis* (blue) using *Candida Panel* and (**b**) *A. fumigatus* (green) and *A. niger* (blue) using *Filamentous Fungi Panel.* (Fluorescent dyes FAM = blue, HEX = green).

**Figure 6 jof-06-00308-f006:**
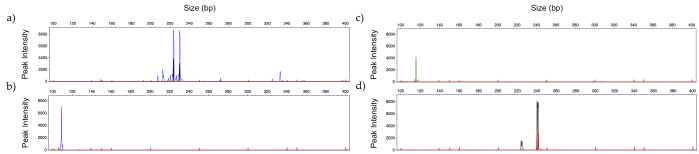
GeneScan profiles obtained by multiplex analysis of DNA extracted from serum spiked with: (**a**) *C. krusei* (blue) and (**b**) *C. albicans* (blue) using *Candida Panel*; and for (**c**) *A. fumigatus* (green) and (**b**) *R. arrhizus* (black) using *Filamentous Fungi Panel*. (Fluorescent dyes FAM = blue, HEX = green, NED = black).

**Figure 7 jof-06-00308-f007:**
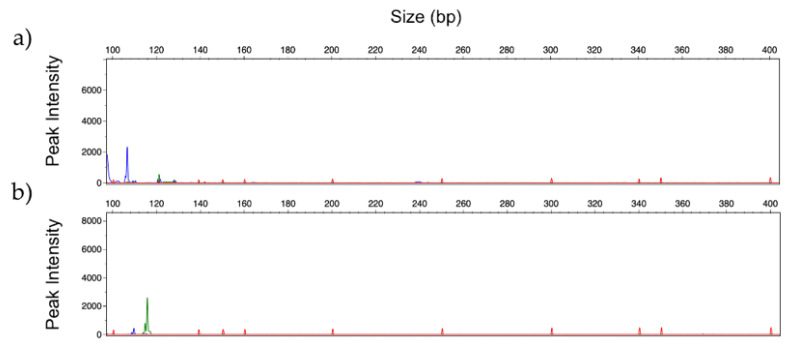
GeneScan profiles obtained by multiplex analysis of clinical samples identified as: (**a**) *C. albicans* (blue) and (**b**) *A. fumigatus* (green), using *Candida Panel* and *Filamentous Fungi Panel*, respectively. (Fluorescent dyes FAM = blue, HEX = green).

**Table 1 jof-06-00308-t001:** Sequences and dye of the primers selected. These primers are included in the Portuguese Patent n° PT 115216.

*Candida Panel*	*Filamentous Fungi Panel*				
Species	Primer Sequence (5′ to 3′)	Dye Label	Species	Primer Sequence (5′ to 3′)	Dye Label
*C. albicans*	F-ttggaatcacttcaccagga R-tttccgtggcatcagtatca	FAM	*A. fumigatus*	F-gccctcttccgttattcctt R-gcgcattgatagctacctcaggc	HEX
*C. glabrata*	F-acacctacgagaaaccaaca R-tagcggtcatccagcatca	NED	*A. flavus*	F-gggatcgacactcggactt R-ctggtaagagcttgtgggtg	HEX
*C. krusei*	F-acagcagtcgcaggccc R-gtcggagacataaccgc	FAM	*A. niger*	F-ccctccttccaaacaaacaa R-tccagatcggctacacagaa	FAM
*C. parapsilosis*	F-aaagtgctacacacgcatcg R-ggcttgcaatttcatttcct	HEX	*A. terreus*	F-gcggatgcaaggtgtaattt R-tactgcgcgttagttgaagc	NED
*C. tropicalis*	F- ccccaccaaaaacatacatacat R-ttacattcagcccgccacag	FAM	*R. arrhizus*	F-agaagcaaaatcatcgtcgaaag R-cgtaggtccagcgtaaacttg	NED

**Table 2 jof-06-00308-t002:** List of species tested and results obtained by multiplex analysis. CA—Candida albicans; CP. Candida parapsilosis; CG—Candida glabrata; CT—Candida tropicalis; CK—Candida krusei; AF—Aspergillus fumigatus, AFL—Aspergillus flavus; NA—Aspergillus niger; AT—Aspergillus terreus; RA—Rhizopus arrhizus.

Species	Isolates Tested	No. of Isolates Amplified with Each Primer Pair										
		CA	CP	CG	CT	CK	AF	AFL	AN	AT	RA	Absence of Amplification
*Candida albicans*	21	21	-	-	-	-	-	-	-	-	-	-
*Candida parapsilosis*	14	-	14	-	-	-	-	-	-	-	-	-
*Candida glabrata*	20	-	-	20	-	-	-	-	-	-	-	-
*Candida tropicalis*	7	-	-	-	7	-	-	-	-	-	-	-
*Candida krusei*	5	-	-	-		5	-	-	-	-	-	-
*Candida bracarensis*	4	-	-	-	-	-	-	-	-	-	-	4
*Candida metapsilosis*	5	-	-	-	-	-	-	-	-	-	-	5
*Candida orthopsilosis*	5	-	-	-	-	-	-	-	-	-	-	5
*Candida guilliermondii*	1	-	-	-	-	-	-	-	-	-	-	1
*Candida lusitaniae*	1	-	-	-	-	-	-	-	-	-	-	1
*Loderomyces elongisporus*	1	-	-	-	-	-	-	-	-	-	-	1
*Aspergillus fumigatus*	2	-	-	-	-	-	2	-	-	-	-	-
*Aspergillus flavus*	8	-	-	-	-	-	-	8	-	-	-	-
*Aspergillus terreus*	4	-	-	-	-	-	-	-	-	4	-	-
*Aspergillus niger*	6	-	-	-	-	-	-	-	6	-	-	-
*Aspergillus versicolor*	3	-	-	-	-	-	-	-	-	-	-	3
*Aspergillus unguis*	1	-	-	-	-	-	-	-	-	-	-	1
*Aspergillus westerdijkiae*	1	-	-	-	-	-	-	-	-	-	-	1
*Aspergillus tamarii*	1	-	-	-	-	-	-	-	-	-	-	1
*Aspergillus wentii*	1	-	-	-	-	-	-	-	-	-	-	1
*Aspergillus nidulans*	3	-	-	-	-	-	-	-	-	-	-	3
*Aspergillus tubingensis*	1	-	-	-	-	-	-	-	-	-	-	1
*Rhizopus arrhizus*	3	-	-	-	-	-	-	-	-	-	3	-
*Rhizopus microsporus*	1	-	-	-	-	-	-	-	-	-	-	1
*Lichtheimia corymbifera*	1	-	-	-	-	-	-	-	-	-	-	1
*Mucor* sp.	1	-	-	-	-	-	-	-	-	-	-	1
TOTAL	121	21	14	20	7	5	2	8	6	4	3	28

**Table 3 jof-06-00308-t003:** Results obtained with 35, 40, 45 and 50 amplification cycles using different concentrations of DNA of ten different species, with the respective panels. - means absence of amplification and +++ high, ++ medium and + low intensity.

*Candida Panel*						*Filamentous Fungi Panel*					
Species	DNA	Amplification Cycles				Species	DNA	Amplification Cycles			
		35	40	45	50			35	40	45	50
*C. albicans*	1 ng	+++	+++	+++	+++	*A. fumigatus*	1 ng	+++	+++	+++	+++
	100 pg	++	++	++	++		100 pg	++	++	+++	+++
	10 pg	-	+	++	++		10 pg	-	+	++	++
	1 pg	-	-	+	+		1 pg	-	+	+	+
*C. glabrata*	1 ng	+++	+++	+++	+++	*A. flavus*	1 ng	+++	+++	+++	+++
	100 pg	+++	+++	+++	+++		100 pg	+++	+++	+++	+++
	10 pg	-	++	+++	+++		10 pg	-	+	+	+
	1 pg	-	+	++	++		1 pg	-	-	-	-
*C. parapsilosis*	1 ng	++	++	++	++	*A. niger*	1 ng	+++	+++	+++	+++
	100 pg	++	++	++	++		100 pg	+++	+++	+++	+++
	10 pg	-	+	+	+		10 pg	-	+	++	++
	1 pg	-	-	-	-		1 pg	-	-	+	+
*C. tropicalis*	1 ng	++	++	++	++	*A. terreus*	1 ng	+++	+++	+++	+++
	100 pg	++	++	++	++		100 pg	++	+++	+++	+++
	10 pg	-	+	+	+		10 pg	-	-	+	+
	1 pg	-	-	-	-		1 pg	-	-	-	-
*C. krusei*	1 ng	+++	+++	+++	+++	*R. arrhizus*	1 ng	+++	+++	+++	+++
	100 pg	+++	+++	+++	+++		100 pg	+++	+++	+++	+++
	10 pg	-	+	++	++		10 pg	-	+	++	++
	1 pg	-	+	++	++		1 pg	-	-	-	-

**Table 4 jof-06-00308-t004:** Clinical samples tested in this study with the correspondent multiplex panels. Their origin, previous identification by quantitative PCR (qPCR) and results obtained using both panels are indicated. - means absence of amplification.

Samples		EORTC/MSG Classification	Previous Identification (qPCR)	Multiplex PCR Results	
Code	Origin			*Candida Panel*	*Filamentous* *Fungi Panel*
CNM1	Vitreous humor	Probable	*C. albicans*	*C. albicans*	-
CNM2	CSF	Probable	*C. albicans*	*C. albicans*	-
CNM3	Broncho-aspirate	Probable	*A. fumigatus*	-	-
CNM4	Liver biopsy	Probable	*C. parapsilosis*	*C. parapsilosis*	-
CNM5	Liver biopsy	Probable	*C. albicans*	*C. albicans*	-
CNM6	BAL	Probable	*C. albicans*	*C. albicans*	-
CNM7	CSF	Proven	*C. albicans*	*C. albicans*	-
CNM8	Biopsy	Proven	*C. albicans*	*C. albicans*	-
CNM9	BAL	Probable	*A. fumigatus*	-	*A. fumigatus*
CNM10	BAL	Probable	*A. fumigatus*	-	-
CNM11	BAL	Probable	*C. krusei*	*C. krusei*	-
CNM12	Biopsy	Probable	*C. tropicalis*	*C. tropicalis*	-
CNM13	Biopsy	Probable	*C. glabrata*	-	-
CNM14	BAL	Probable	*C. parapsilosis*	-	-
UHW1	BAL	NA	*A. fumigatus*	-	Inconclusive
UHW2	BAL	NA	*A. fumigatus*	-	Inconclusive
UHW3	BAL	NA	*A. fumigatus*	-	Inconclusive
UHW4	BAL	NA	*A. fumigatus*	-	Inconclusive
UHW5	BAL	NA	*A. fumigatus*	-	Inconclusive
**Controls**					
CNM15	Pulmonary biopsy	NA	*C. guilliermondii*	-	-
CNM16	CSF	NA	*T. asahii*	-	-
CNM17	Biopsy	NA	*L. corymbifera*	-	-
CNM18	Intestinal biopsy	NA	*Rhizopus sp.*	-	-
CNM19	BAL	NA	*P. jiroveccii*	-	-
CNM20	BAL	Not classified	Neg. controls	-	-
CNM21	Broncho-aspirate	Not classified	Neg. controls	-	-
CNM22	Pulmonary biopsy	Not classified	Neg. controls	-	-
CNM23	Whole blood	Not classified	Neg. controls	-	-
CNM24	Hepatic biopsy	Not classified	Neg. controls	-	-

NA—information not available; CNM—National Centre of Microbiology; UHW—University Hospital of Würzburg; CSF—cerebrospinal fluid; BAL—bronchoalveolar lavage.

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
