# Peer review of "Multiplex PCR Based Strategy for Detection of Fungal Pathogen DNA in Patients with Suspected Invasive Fungal Infections"

_jof, 2020, doi:10.3390/jof6040308_

Round 1

Reviewer 1 Report

      In this paper, the authors provided a new multiplex PCR method to detect and identify the common fungal pathogens in patients with IFI. Overall, the manuscript is well organized. But there are still some issues as follows:

  1. Primer pairs used in this study should be listed and explained. Species specific primers were crucial in this method, so the sequences and molecular targets of the primers used in the study should be listed and explained.
  2. Line 16: Was the “IFI” short for “invasive infections” or “invasive fungal infections”?
  3. There were some other methods based on multiplex PCR in fungi detection, these studies should be reviewed in the “Introduction” section.
  4. Line 159 and line 167: The two sentences were repeated, just keeping one is ok.
  5. Why different DNA polymerases were used in the singleplex PCRs and multiplex PCRs in 2.5 section?
  6. I think “ng” and “pg” are not units of concentration, so it’s better to change them in 3.3 section.
  7. In 3.6 section, the qPCR method was used as a contrast, so compared with qPCR, what are the advantages of the new method?
  8. The results showed that the method was not suitable for bronchoalveolar lavages and bronchoaspirates analysis, the reasons for this should be further analyzed in discussion section.

Author Response

We thank Reviewer 1 for these comments and the opportunity to improve the manuscript.

Comments and Suggestions for Authors

In this paper, the authors provided a new multiplex PCR method to detect and identify the common fungal pathogens in patients with IFI. Overall, the manuscript is well organized. But there are still some issues as follows:

Point 1: Primer pairs used in this study should be listed and explained. Species specific primers were crucial in this method, so the sequences and molecular targets of the primers used in the study should be listed and explained.

Response 1: Yes, a great time studying regions and designing primers was spend in this study. Several molecular targets were tested and different non-coding regions were selected. We did not include the sequences of the primers or molecular targets on the manuscript initially because they were included in a National Patent (115216). The reviewers can access all that information at the website https://servicosonline.inpi.pt/pesquisas/main/patentes.jsp?lang=PT

However, now that the patent is published, we can include the primers in the manuscript. Table 1 was included with the primers sequence.

Point 2: Line 16: Was the “IFI” short for “invasive infections” or “invasive fungal infections”?

Response 2: IFI was the short of invasive fungal infections. The missing word “fungal” was included in line 16.

Point 3: There were some other methods based on multiplex PCR in fungi detection, these studies should be reviewed in the “Introduction” section.

Response 3: In the introduction section we included multiplex assays that are commercially available. To answer the reviewer, we include two other described methods and two revision studies that present many other PCR multiplex methods.

Point 4: Line 159 and line 167: The two sentences were repeated, just keeping one is ok.

Response 4: Sentence from line 159 was deleted.

Point 5: Why different DNA polymerases were used in the singleplex PCRs and multiplex PCRs in 2.5 section?

Response 5: PhusionTM High-Fidelity DNA Polymerase is a fast and accurate high-fidelity DNA polymerase that was suitable for all optimization singleplex PCR reactions, while PlatinumTM II Taq Hot-Start DNA polymerase was recommended as more suitable for the multiplex reactions.

Point 6: I think “ng” and “pg” are not units of concentration, so it’s better to change them in 3.3 section.

Response 6: This analysis was performed to teste if the multiplex PCR was able to amplify in the presence of those amounts of DNA. Indeed, ng and pg are not units of concentration but the amount of DNA present in the PCR tube. To clarify, that information was included in the text.

Point 7: In 3.6 section, the qPCR method was used as a contrast, so compared with qPCR, what are the advantages of the new method?

Response 7: We believe that, in comparison with qPCR, the possibility of observing the amplification peak in the correct region of the panel is an advantage of the new method because it helps roll out inconclusive amplifications.

Point 8: The results showed that the method was not suitable for bronchoalveolar lavages and bronchoaspirates analysis, the reasons for this should be further analyzed in discussion section.

Response 8: Unlike other methods that are based on multicopy DNA regions, our method is based on the specific amplification of different non-coding regions for each species. So, we believe that the main reason for our results with bronchoalveolar lavages is indeed the dilution of the samples. This concern is included in the discussion section lines 361-364.

Reviewer 2 Report

The objective of this study is to establish a sensitive and specific multiplex PCR strategy for the identification of the most clinically important pathogenic species of Candida spp., Aspergillus spp. and Rhizopus arrhizus. The study has a very interesting proposal and results, however the text is not well written and needs a substantial review.

-Lines 15-16: Replace “A new and easy to interpret PCR multiplex strategy, for the identification of the most common fungal species involved in invasive infections (IFI) was developed in this work” by “A new and easy PCR multiplex strategy, for the identification of the most common fungal species involved in invasive infections (IFI) was developed in this work”.

-Lines 21-25: The following sentence needs to be rewritten: “This method was suitable for samples from sterile sites, with a sensitivity of 89% and specificity of 100%. This study showed that the method is suitable for the identification of the ten most important fungal species involved in IFI, not only from positive blood cultures but also from clinical samples from sterile sites, and mixed infections”. These two phrases are repetitive and confuse.  

-Lines 25-27: Replace “The possibility of the user to see the peak in the specific region of the panel, with the correct fluorescence dye, is a unique characteristic of the assay that aids ruling-out unspecific amplifications” by “The method provides an unique characteristic of seeing the peak in the specific region of the panel with the correct fluorescence dye, that aids ruling-out unspecific amplifications.

-Line 35: Do the authors want to say “Actually” or “Currently”?

-Line 42: “Notably, the authors also reported that their…”. The “authors” should be specified and the data better connected with the previous phrase.

-Lines 54-59: Many parts of the text needs to be rewritten to become more objective and concise, for example: “To enhance survival of infected patients, an early and adequate identification of the infecting species is crucial, since it allows the selection of appropriate antifungal treatment, as different species differ in their susceptibility to antifungal agents [11,12]. Moreover, the fact that the risk of developing deep organ involvement, and the severity of clinical manifestations differs depending on the infecting species, the significance of a correct diagnosis is very high [2]”. The text is repetitive and needs to be better organized.

-Line 71: Replace “However, blood cultures are relatively inexpensive, use widely available/accepted technologies..” by ““However, blood cultures are relatively inexpensive, require available and accepted technologies..”.

-Lines 73-91: The text needs to be rewritten and summarized in only one paragraph. Be more specific and concise.

-Line 106: Write Candida spp., Aspergillus spp., and not sp.

-Why Candida auris was not included in the Candida Panel? Comments about this new emerging species should be added in the discussion section.

-Lines 125-137: I suggest include a Table summarizing the main information about strains, for example number of strains for each species, perceptual rate of clinical and environmental sources, etc.

-Line 137: Add more details in this phrase: “Human Genomic DNA (Promega) was also tested”.

-Lines 142-146: The methodologies for yeast DNA extraction should be better described. Also, explain in the text the reasons to use two different methodologies.

-Lines 159-160 and 167-168: The phrase “All manipulations were performed in a category 2 laminar flow cabinet to prevent contamination” is written twice in the same paragraph. Also, this information is repeated in other paragraphs.

-Lines 195-197: “These samples included 19 samples of patients with probable/proven fungal infection and 10 negative controls, and were obtained from biopsies, bronchoalveolar lavages (BAL), cerebrospinal fluid, and vitreous humor”. Why did the authors include samples from patients with probable fungal infection from DNA collection? To validate the developed method is important to analyze samples with confirmed diagnostic (positive and negative, but not “Probable”).

-Figures 2-7: The numbers in the graphs are very small. It is not readable.

-Table 1: Write in the legend the mean of CA, CP, etc. For example, CA: Candida albicans.

-Lines 237-238: “The same multiplex PCR conditions were also tested using a protocol for colony PCR [29] showing similar results (data not shown)”. These results are presented in the reference 29? This phrase is confused. Also, it is not connected with the text.

-Lines 314-321: This is a very important comments about the results obtained, however they were not discussed. Add a paragraph about it in the discussion section.

-In general, the discussion section needs to be improved. The results were not compared with other studies. Several references about multiplex PCR for fungal identification can be included.

Author Response

The authors thank reviewer 2 for the comments and appreciate the opportunity to improve the manuscript.

Comments and Suggestions for Authors

The objective of this study is to establish a sensitive and specific multiplex PCR strategy for the identification of the most clinically important pathogenic species of Candida spp., Aspergillus spp. and Rhizopus arrhizus. The study has a very interesting proposal and results, however the text is not well written and needs a substantial review.

Point 1: Lines 15-16: Replace “A new and easy to interpret PCR multiplex strategy, for the identification of the most common fungal species involved in invasive infections (IFI) was developed in this work” by “A new and easy PCR multiplex strategy, for the identification of the most common fungal species involved in invasive infections (IFI) was developed in this work”.

Response 1: The sentence was replaced as suggested.

Point 2: Lines 21-25: The following sentence needs to be rewritten: “This method was suitable for samples from sterile sites, with a sensitivity of 89% and specificity of 100%. This study showed that the method is suitable for the identification of the ten most important fungal species involved in IFI, not only from positive blood cultures but also from clinical samples from sterile sites, and mixed infections”. These two phrases are repetitive and confuse. 

Response 2: The sentences were rewritten for a better reading.

Point 3: Lines 25-27: Replace “The possibility of the user to see the peak in the specific region of the panel, with the correct fluorescence dye, is a unique characteristic of the assay that aids ruling-out unspecific amplifications” by “The method provides an unique characteristic of seeing the peak in the specific region of the panel with the correct fluorescence dye, that aids ruling-out unspecific amplifications.

Response 3: The sentence was replaced as suggested.

Point 4: Line 35: Do the authors want to say “Actually” or “Currently”?

Response 4: Neither Actually or Currently, “Actually” is wrongly placed, so it was removed from the sentence.

Point 5: Line 42: “Notably, the authors also reported that their…”. The “authors” should be specified and the data better connected with the previous phrase.

Response 5: The sentences were rewritten for a better reading and connection.

Point 6: Lines 54-59: Many parts of the text needs to be rewritten to become more objective and concise, for example: “To enhance survival of infected patients, an early and adequate identification of the infecting species is crucial, since it allows the selection of appropriate antifungal treatment, as different species differ in their susceptibility to antifungal agents [11,12]. Moreover, the fact that the risk of developing deep organ involvement, and the severity of clinical manifestations differs depending on the infecting species, the significance of a correct diagnosis is very high [2]”. The text is repetitive and needs to be better organized.

Response 6: These sentences were organized, as suggested.

Point 7: Line 71: Replace “However, blood cultures are relatively inexpensive, use widely available/accepted technologies..” by ““However, blood cultures are relatively inexpensive, require available and accepted technologies..”.

Response 7: The sentence was organized, as suggested.

Point 8: Lines 73-91: The text needs to be rewritten and summarized in only one paragraph. Be more specific and concise.

Response 8: The sentence was organized, as suggested.

Point 9: Line 106: Write Candida spp., Aspergillus spp., and not sp.

Response 9: The correction was performed as indicated.

Point 10: Why Candida auris was not included in the Candida Panel? Comments about this new emerging species should be added in the discussion section.

Response 10: When this study was planned, Candida auris was not yet described as a serious problem. The development of this method has taken us a long time to test all species and adjust the panels. Candida auris is now being included in the panel. We have already the species-specific primers design, and we are testing them.

Point 11: Lines 125-137: I suggest include a Table summarizing the main information about strains, for example number of strains for each species, perceptual rate of clinical and environmental sources, etc.

Response 11: The number of strains for each species is indicated in pervious table 1. Concerning the percentual of clinical and environmental strains that information was include in the text.

Point 12: Line 137: Add more details in this phrase: “Human Genomic DNA (Promega) was also tested”.

Response 12: The sentence was changed to “Human Genomic DNA (Promega) was also used in the development of panels to assess cross-amplification” to explain that this DNA was used to test if any cross-amplification would occur with the primers design that would hamper the identification.

Point 13: Lines 142-146: The methodologies for yeast DNA extraction should be better described. Also, explain in the text the reasons to use two different methodologies.

Response 13: For yeast DNA extraction we used the commercial kit JetQuick TM DNA Purification Kit from Invitrogen as followed the Kit’ instructions. Colony-PCR, was used to test if we could use this methodology directly with lysed fungal cells, without the need to extract DNA. This information was included in the manuscript.

Point 14: Lines 159-160 and 167-168: The phrase “All manipulations were performed in a category 2 laminar flow cabinet to prevent contamination” is written twice in the same paragraph. Also, this information is repeated in other paragraphs.

Response 14: This sentence was removed from lines 159-160 and 142-143, as suggested.

Point 15: Lines 195-197: “These samples included 19 samples of patients with probable/proven fungal infection and 10 negative controls, and were obtained from biopsies, bronchoalveolar lavages (BAL), cerebrospinal fluid, and vitreous humor”. Why did the authors include samples from patients with probable fungal infection from DNA collection? To validate the developed method is important to analyze samples with confirmed diagnostic (positive and negative, but not “Probable”).

Response 15: The samples, from the DNA collection, selected for this study were included while stablishing the DNA collection. Our criterium was to select DNA samples that were tested by qPCR. However, in the final checking of the samples while writing the manuscript, the information about classification was “Probable”.

Point 16: Figures 2-7: The numbers in the graphs are very small. It is not readable.

Response 16: The numbers in the figures, refers to the intensity of the fluorescence (Y-axis) and the molecular size of the fragment (X-axis). They are automatically indicated by the GeneScan program. What we did was to put all the images inside each figure with the same scales, so that readers could compare the position, intensity and dye color of the peaks of each Panel. The only way to enhance the numbers would be to increase the figure size. We enhanced the figure sizes.

Point 17: Table 1: Write in the legend the mean of CA, CP, etc. For example, CA: Candida albicans.

Response 17: This information was written in the legend of the table, as suggested.

Point 18: Lines 237-238: “The same multiplex PCR conditions were also tested using a protocol for colony PCR [29] showing similar results (data not shown)”. These results are presented in the reference 29? This phrase is confused. Also, it is not connected with the text.

Response 18: Reference 29 describes the colony PCR protocol. The reference was removed (it is already indicated in material and methods) and the two sentences connected to facilitate understanding.

Point 19: Lines 314-321: This is a very important comments about the results obtained, however they were not discussed. Add a paragraph about it in the discussion section.

Response 19: A paragraph discussing the sensitivity and specificity was included in the discussion section.

Point 20: In general, the discussion section needs to be improved. The results were not compared with other studies. Several references about multiplex PCR for fungal identification can be included.

Response 20: The discussion section was improved to include discussion and comparison with other multiplex studies.